

# High definition video loggers provide new insights into behaviour, physiology, and the oceanic habitat of a marine predator, the yellow-eyed penguin

Thomas Mattern[1,2], Michael D. McPherson[3], Ursula Ellenberg[2,4], Yolanda van Heezik[1] and Philipp J. Seddon[1]

[1] Department of Zoology, University of Otago, Dunedin, Otago, New Zealand
[2] Global Penguin Society, Puerto Madryn, Chubut, Argentina
[3] CTNova Ltd., Leighton Buzzard, Bedfordshire, United Kingdom
[4] Department of Ecology, Environment and Evolution, La Trobe University, Melbourne, Victoria, Australia

## ABSTRACT

Camera loggers are increasingly used to examine behavioural aspects of free-ranging animals. However, often video loggers are deployed with a focus on specific behavioural traits utilizing small cameras with a limited field of view, poor light performance and video quality. Yet rapid developments in consumer electronics provide new devices with much improved visual data allowing a wider scope for studies employing this novel methodology. We developed a camera logger that records full HD video through a wide-angle lens, providing high resolution footage with a greater field of view than other camera loggers. The main goal was to assess the suitability of this type of camera for the analysis of various aspects of the foraging ecology of a marine predator, the yellow-eyed penguin in New Zealand. Frame-by-frame analysis allowed accurate timing of prey pursuits and time spent over certain seafloor types. The recorded video footage showed that prey species were associated with certain seafloor types, revealed different predator evasion strategies by benthic fishes, and highlighted varying energetic consequences for penguins pursuing certain types of prey. Other aspects that could be analysed were the timing of breathing intervals between dives and observe exhalation events during prey pursuits, a previously undescribed behaviour. Screen overlays facilitated analysis of flipper angles and beat frequencies throughout various stages of the dive cycle. Flipper movement analysis confirmed decreasing effort during descent phases as the bird gained depth, and that ascent was principally passive. Breathing episodes between dives were short ($<1$ s) while the majority of the time was devoted to subsurface scanning with a submerged head. Video data recorded on free-ranging animals not only provide a wealth of information recorded from a single deployment but also necessitate new approaches with regards to analysis of visual data. Here, we demonstrate the diversity of information that can be gleaned from video logger data, if devices with high video resolution and wide field of view are utilized.

Corresponding author
Thomas Mattern,
t.mattern@eudyptes.net

## INTRODUCTION

Examining the at-sea behaviour of marine animals has long been a challenging endeavour. Direct visual observations of behaviour are almost impossible, especially when most of it happens under the ocean's surface. In recent decades, advances in telemetry technologies and the emergence of bio-logging hardware have provided the means to track marine animals and reveal their foraging behaviour in great detail. Starting in the 1970s with rather crude location estimates and limited data quality recorded by unwieldy devices that could only be used on large animals, advancements in micro-electronics have resulted in ever smaller and more accurate loggers to pinpoint an animal's position to within a few metres and record their diving depths with oceanography-grade precision (*Wilmers et al., 2015*). New technologies such as accelerometers and gyroscopes further refined methods to study marine habitat use (e.g., *Noda et al., 2014*). Yet placing dive metrics into a complex behavioural and environmental context can be difficult; ideally, a reference framework based on direct observations is used to match up dive metrics and actual behaviours (e.g., *Moreau et al., 2009*; *Volpov et al., 2016*). So, the original dilemma of having to make direct observations of marine animal behaviours still persists. Animal-borne video recorders offer the means to overcome this problem.

In recent years, animal-borne camera systems have made it possible to log in situ observations of behaviour from the animal's point of view (*Moll et al., 2007*). For example, deployment of lightweight video cameras on flying birds provided new perspectives on prey pursuit in falcons (*Kane & Zamani, 2014*) and revealed how albatrosses use the presence of killer whales to locate prey (*Sakamoto et al., 2009*). No other animal group has been more subject to deployment of video recording devices in recent years than marine animals. By overcoming the observational barrier at sea, video loggers are providing copious amounts of novel data that range from identification of feeding strategies (*Ponganis et al., 2000*; *Takahashi et al., 2008*) and previously unknown food sources (*Thiebot et al., 2017*), to social interactions such as group foraging (*Sutton, Hoskins & Arnould, 2015*) or kleptoparasitism (*Handley & Pistorius, 2015*). Video data also offer the means to calibrate other bio-logging data (*Watanabe & Takahashi, 2013*; *Gómez-Laich et al., 2015*).

What most of these studies have in common is their focus on specific behavioural traits while providing limited information about the environment the behaviours occurred in. This is principally due to limitations of the video hardware used, which has to be small and lightweight so as to not overly impede the study animal's movement capabilities (*Ludynia et al., 2012*) and hence behaviour. As a result, video quality (i.e., image resolution and field of view/FOV) is sacrificed in favour of smaller cameras (e.g., *Watanabe & Takahashi, 2013*; *Gómez-Laich et al., 2015*; *Thiebot et al., 2016*; *Thiebot et al., 2017*). However, with the rise in popularity of action cams on the consumer market, new video devices have recently become available with high definition video capabilities and wide-angle optics, suitable for deployment even on smaller marine animals such as penguins. This leap in quality has significant implications for the study of marine animals as it not only allows more accurate monitoring of a wide-range of aspects of behaviours such as specific pursuit strategies and capture efficiency, as well as prey identification and interactions with other species, but

also provides new opportunities for the visual analysis of the environment the animals use. This is particularly relevant in species that forage at the seafloor where video data can provide extensive information about the benthic habitat (*Watanuki et al., 2008*).

The yellow-eyed penguin (*Megadyptes antipodes*) in New Zealand is known to be a benthic forager (*Mattern et al., 2007*) that feeds primarily on demersal fish species (*van Heezik & Davis, 1990*; *Moore et al., 1995*). It has been suggested that this strategy might come at the expense of reduced behavioural flexibility, with subsequent vulnerability to changes in the marine environment (*Mattern et al., 2007*). In particular, degradation of seafloor ecosystems in the wake of commercial bottom fisheries are suspected to influence yellow-eyed penguin foraging success and population developments (*Browne et al., 2011*; *Mattern et al., 2013*). While the species' at-sea movement and diving behaviour have been subject to a number of studies in the past decades (*Moore et al., 1995*; *Mattern et al., 2007*; *Mattern et al., 2013*), information about their benthic habitat is very limited.

To assess the extent to which penguin behaviour and foraging success correlate with the composition of the benthic habitat, we developed a camera logger that records full high-definition (HD) videos through wide-angle lenses. The main focus of our study was to assess the suitability of the device for the visual analysis of penguin prey pursuit behaviour and characteristics of the benthic ecosystem. However, the deployment revealed far more information than was anticipated. The video data provided novel insights into physiological aspects of the penguin's diving activities and allowed us to draw conclusions about prey capture techniques. In this paper, we summarise our findings, demonstrate analytical approaches to evaluate animal-borne video data, and highlight the multi-disciplinary potential of wide-angle, full HD video loggers.

## MATERIALS AND METHODS

### Study site and species

The yellow-eyed penguin, classified as ''Endangered'' by the IUCN Redlist (*BirdLife International, 2016*), is one of five penguin species endemic to the New Zealand region and occurs on the sub-Antarctic Auckland and Campbell Islands as well as the south-eastern coastlines of New Zealand's South Island and Stewart Island (*Seddon, Ellenberg & van Heezik, 2013*). This study was carried out at the Boulder Beach complex, Otago Peninsula, South Island, New Zealand (45.90°S, 170.56°E). Penguins from this site have been subject to foraging studies that have suggested substantial impact of bottom trawling activities on the yellow-eyed penguins' at-sea movements (*Ellenberg & Mattern, 2012*; *Mattern et al., 2013*).

### Video logger & deployment

We developed a high-definition video logger (dimensions LxWxH, $89 \times 41 \times 21$ mm; weight: 78 g) which is combined with a time-depth recorder (TDR, $31 \times 12 \times 11$ mm, 6.5 g; AXY-depth, Technosmart Ltd. Italy) and a GPS logger (modified, epoxy encased i-gotU, GT-120; Mobile Action Technology Inc., Taipei, Taiwan, $31 \times 22 \times 11$ mm, 12 g). The latter two devices were combined into a single unit by gluing the AXY-depth to the longer side of the GPS device. Camera and logger combination were then attached individually in

line to the lower back of the penguin using adhesive tape (*Wilson et al., 1997*). Additional drag of the devices was principally limited to the camera's frontal area (*Bannasch, Wilson & Culik, 1994*).

The camera logger consisted of a modified Mobius action-cam with a 130° wide-angle lens (http://www.mobius-actioncam.com). To achieve the smallest and lightest device possible, the camera electronics, video sensor and lens were removed from the casing and the battery replaced with a 1,200 mAh Lithium Polymer battery to extend recording time. A small bespoke timer board was developed to allow the camera to be fired at a pre-determined time. Connections were provided to allow programming logger start time and also to access the camera's USB port for managing camera setting, extracting the video data and recharging the battery. The board was isolated electrically to prevent the contacts from shorting as sea-water is conductive. Activation of the interface was achieved using a Hall-effect device. An Arduino-based interface was developed to allow the current date/time and logger start time to be set. The camera was programmed to record video data at a resolution of 1,920 × 1,080 pixels (1080p) at a frame rate of 30 frames per second. Video data were recorded in H.264 MPEG4 format and stored on a 32 GB MicroSD card. The camera was programmed to start recording at 11 am the following day when it was assumed that the penguin had completed its travel phase and arrived at its foraging destination. The camera operated from the programmed start time until the battery fell below the minimum operating voltage of the camera (ca. 2–4 h). The device was recovered when the penguin returned from its foraging trip; data were then downloaded through the camera's USB interface.

Since the logger stores video data as a series of full frame images ('progressive scan'), it was possible to conduct a frame-by-frame analysis to accurately time components of the bird's behaviour—i.e., breathing intervals, flipper beat frequencies and amplitudes—as well as time spent over certain benthic habitats. Video analysis was conducted in professional editing software (Adobe Premiere Pro CS 6, Adobe Systems Inc., San Jose, CA, USA) which allows the quick and precise backward and forward navigation of the video material using the keyboard ("scrubbing") and provides the option to display frame number in the preview timer.

The video logger was deployed on a breeding male yellow-eyed penguin tending two chicks on 17 December 2015. Deployment occurred at the penguin's nest on the evening of 17 December. The bird was removed from the nest and placed in a cloth bag to reduce stress. The instrumentation procedure lasted around 20 min after which the penguin was released back on its nest. The bird left on a single foraging trip on 18 December before the device was recovered on 19 December; the penguin continued to breed normally after the deployment.

## Failure to record GPS data

Upon device recovery, it became apparent that the GPS logger did not record any data after the camera had started operating. It has since become evident that the Mobius action-cam generates significant electromagnetic interference which prevented the GPS logger from

functioning properly. This can be rectified by wrapping the camera with electrical shielding tape; however, in our case the lack of shielding resulted in failure to record GPS data.

## Analysis of behaviours & habitat

For detailed analysis of behaviours, we randomly selected 12 dive cycles (dive cycle = beginning of the surface period until the end of the following dive's ascent phase) from the 46 dive cycles recorded (i.e., one-fourth of all dives) independent from prevalent behaviours exhibited during these dives. This was due to the labour-intensive frame-by-frame analysis necessary for several of the behaviours. Future analyses are ideally conducted automatically using machine learning algorithms to reduce analysis time and increase accuracy (e.g., *Valletta et al., 2017*).

### Prey pursuits & capture

We defined the beginning of a prey pursuit as the moment when the penguin markedly accelerated while swimming along the seafloor; the end was reached when the penguin decelerated again to its previous cruise speed (if no prey was caught), or when the prey item was swallowed completely. Acceleration and deceleration were associated with temporary blurring of the video footage due to irregular body movement, allowing for exact timing of prey pursuits. Where possible, prey species were identified from frames providing a clear view of the prey item.

### Benthic habitat

For all dives, the benthic habitat was classified according to sediment type (fine sand, coarse sand with shell fish fragments, gravel), sediment structure (flat, sediment ripples) and composition of the epibenthic communities. For the latter, we used a presence/absence approach with easy-to-identify epibenthic species brittlestars (*Ophiuroidea*), anthozoans (anemones and soft corals), and horse mussels (*Atrina zelandica*), within a 30-frame time window. Figure 1 provides a photographic overview of the different habitat characteristics used. Future deployments with a functional GPS logger can be used for more elaborate analysis of the benthic habitat, e.g., the creation of biodiversity indices.

Beyond prey and habitat interactions, the video data offered the opportunities to analyse various physiological aspects of the penguin's behaviour.

### Flipper movements

During dives, flipper beat frequencies (beats per minute, BPM) were determined by counting the number of frames required to complete one flipper beat cycle, beginning the count when the flipper angle reached its maximum upward inclination and ending with the frame prior to the subsequent maximum upward inclination. In the video editing software, we overlaid a template indicating 10, 30, 50, 70 and 90-degree angles radiating from the base of the flippers on the video data (https://vimeo.com/179414575). This allowed us to visually determine maximum amplitude of each flipper beat to the nearest 5°.

### Surface breathing & underwater exhalation

We timed breathing events when the penguin was at the surface following a dive. Noting frame numbers when the bird raised its head out of the water before lowering below the

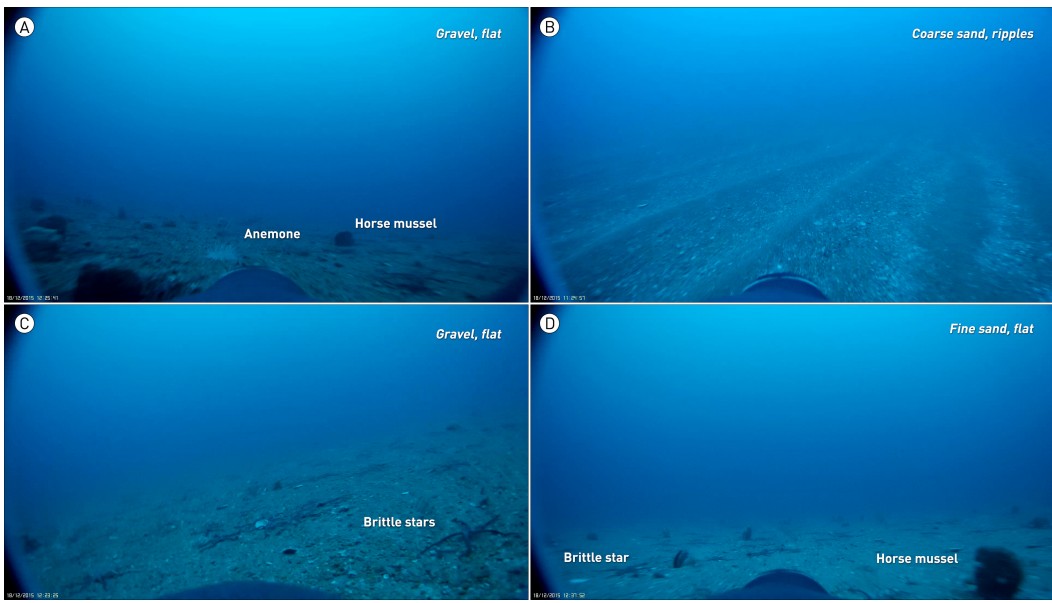

**Figure 1  Types of benthic habitat utilized by a yellow-eyed penguin fitted with wide-angle, full HD camera logger.** (A) Flat gravel sediment with a diverse epibenthic community of horse mussels, sponges, ascidians and anemones; a habitat preferred by blue cod. (B) Coarse sand with pronounced ripples, typical habitat where opalfish were caught. (C) Flat gravel sediment with almost no sessile epibenthos; the presence of scavenging brittle stars indicates bottom fishing disturbance. (D) Flat, fine sand habitat with horse mussels and few brittle stars. The latter two habitats were also frequented by blue cod.

surface again made it possible to determine the times the penguin was able to respire (https://vimeo.com/179414575#t=145). Additional observations of exhalations during the dive were noted.

A selection of edited video clips demonstrating the various behaviours and habitat types described above can be accessed via https://vimeo.com/album/4103142.

## Dive data analysis

Dive data recorded by the TDR at 1 s intervals and depth resolution of ∼0.1 m were analysed following methods described in detail in *Mattern et al. (2007)*. Dives were classified as pelagic or benthic dives using dive profile characteristics, where near horizontal bottom phases with little vertical variance as well as consistent maximum dive depths on consecutive dives were used as cues for diving along the seafloor. This approach was validated by recorded video data. The TDR also recorded tri-axial accelerometer data which have yet to be analysed.

Statistical analysis was carried out in R 3.4.2 (*R Core Team, 2014*). Correlations were examined as linear models (Pearson's correlation). Comparisons were conducted as simple t-tests accounting for unequal variances (Welch's $t$-Test, *Ruxton, 2006*).

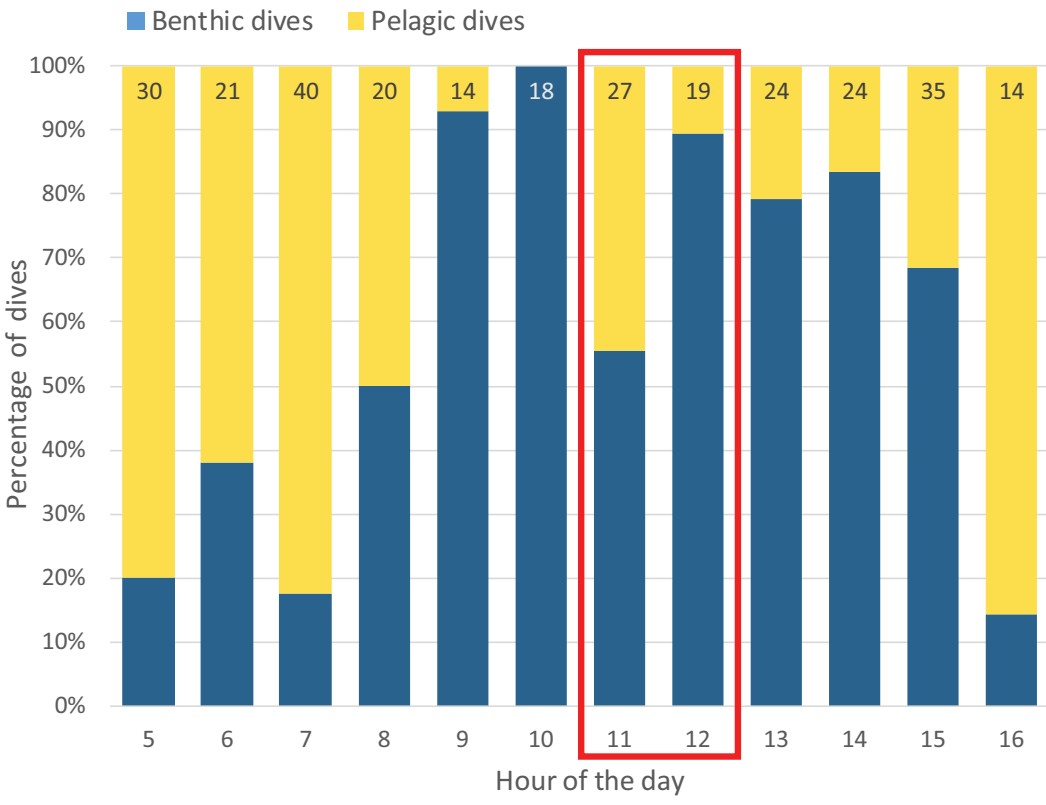

**Figure 2  Proportion of benthic and pelagic dives throughout the yellow-eyed penguin's foraging trip while fitted with a camera logger.** Numbers at the top end of bars indicate number of dives performed during the corresponding hour. Red box indicates the hours during which continuous camera footage was recorded.

## Permits

This study was approved by the Animal Ethics Committee of the University of Otago (UOO AEC 69/15) and field experiments conducted under research permits issued by the New Zealand Department of Conservation (45799-FAU).

## RESULTS

### Foraging trip length, diving events and video coverage

The day following camera deployment, the penguin performed a 10.7 hour-long foraging trip. The first dive event was recorded at 5:30 hrs and the last event concluded at 16:10 hrs. The bird performed 286 dives of which 159 dive profiles matched the criteria for benthic dives (Fig. 2). Median dive depth reached during benthic dives was 54.4 m (range: 4.8–62.1 m, $n = 159$) whereas the majority of pelagic dives occurred in the upper 10 m of the water column (median: 7.8 m, range: 0.5–31.7 m, $n = 127$); camera footage confirmed these to be principally travelling behaviour (https://vimeo.com/179414642). For the first 3.5 h of the foraging trip (05:30–09:00 hrs) the bird performed mainly pelagic dives, indicating primarily travelling behaviour towards its main foraging grounds; yellow-eyed penguins

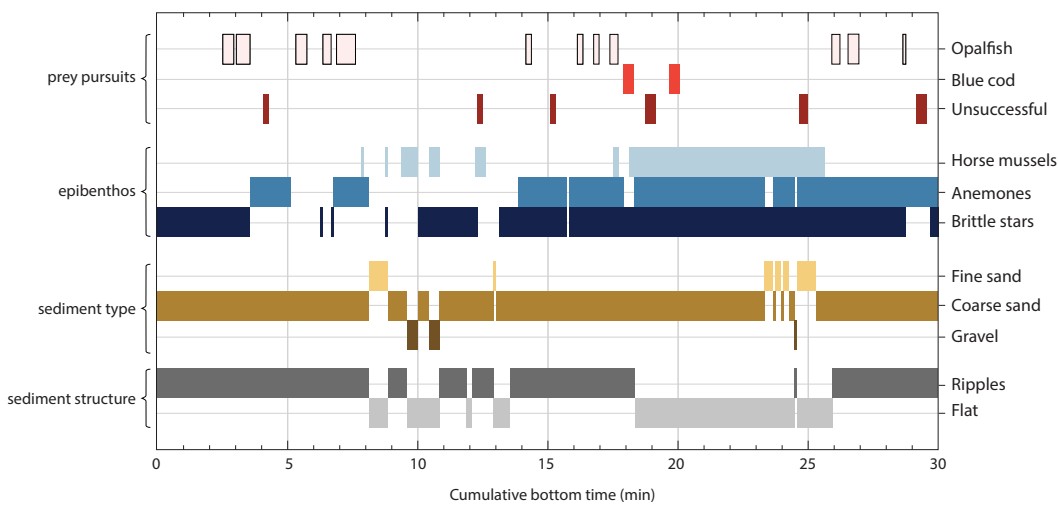

**Figure 3** **Timeline of a yellow-eyed penguin's prey pursuit events in relation to benthic features, i.e., composition of epibenthic community, sediment type and structure.** The $x$-axis indicates the cumulative time the penguin spent at the seafloor (29.9 min) during the 2 h of camera operation. The top three rows of bars indicate frequency and length of prey pursuit events while the remaining rows of bars highlight various features of the seafloor. The length of each bar shows how long the bird foraged over the respective feature.

are known to exhibit high individual site fidelity with regards to foraging locations (*Moore, 1999*; *Mattern et al., 2007*). Between 09:00 and 16:00 hrs the bird principally devoted its time to benthic diving while shallow dives dominated the remaining 10 min of the foraging trip (Fig. 2).

## Video coverage & quality

The camera operated continuously from 11:00:22 hrs to 13:01:43 hrs. Due to frame loss representing a mean 1.6 seconds of footage when video data were written to the file every 3 min, total length of the recorded footage amounted to 2 h 8 s. Forty-six complete dives were video recorded which corresponds to 16% of all dive events; of these 32 dives were benthic dives. However, dives were longer during the middle of the day so that camera footage covered 25% of the trip's cumulative dive time. The video quality proved to be significantly better than that recorded with other animal-borne cameras deployed on penguins to date (https://vimeo.com/268905870). The light sensitivity of the camera was adequate to record clear images at dive depths close to 70 m and, combined with the large field of view, facilitated detailed frame-by-frame analysis.

## Prey pursuits & capture

A total of 20 prey pursuits were recorded at the seafloor (Fig. 3). Fourteen of these resulted in successful capture of either opalfish (*Hemerocoetes monopterygius*, 10 specimens) or blue cod (*Parapercis colias*, 2 specimens); prey species could not be identified during two captures, but the penguin's searching behaviour and ease of ingestion suggested these were opalfish and we include them with opalfish captures below and in Fig. 3. All of these prey pursuits occurred at the sea floor with the penguin swimming very close to the bottom

(https://vimeo.com/179414724). During the camera operation time, the penguin spent 5.7 min on prey pursuit, which corresponds to 19% of the total time the bird foraged along the seafloor (29.9 min) and 6% of its total dive time (89.9 min). The penguin spent most of its active prey pursuit on opalfish (total 3.8 min, 12 events), 0.7 min were used to capture blue cod (2 events), and 1.2 min of prey pursuit did not result in successful prey capture (Fig. 3).

Two main prey pursuit strategies became apparent that were associated with prey species. When catching opalfish, the penguin would glide closely above the seafloor, sometimes briefly accelerating before starting to hover over a certain spot while repeatedly pecking at the substrate until the prey item was captured (https://vimeo.com/179414724). During encounters with blue cod prolonged pursuits ensued during which fish zigzagged at a fast pace along the seafloor (https://vimeo.com/179414724#t=2m46s). In one instance the fish was caught as it appeared to seek shelter at the base of a horse mussel protruding from the substrate (https://vimeo.com/179414724#t=2m55s). An unsuccessful prey pursuit of blue cod ended with the fish escaping under what appeared to be a half-buried back plate of a dishwasher (https://vimeo.com/179414777). A third blue cod encounter occurred just seconds after a successful capture of an opalfish; it seems likely that the resulting prolonged bottom time and oxygen-demanding prey pursuits drove the penguin to carry the fish to the surface at an almost vertical angle as indicated sun disc's central position in the frame; the fish was ultimately dropped at the surface (https://vimeo.com/179414724#t=3m07s).

### Benthic habitat

During the video logger's operating time, the penguin spent 29.9 min foraging along the seafloor. The majority of the penguin's bottom time (90%) was spent over coarse sand, whereas time spent over fine sand (7%) and gravel (0.9%) was negligible (Figs. 1 & 3). Two thirds of the bottom time (65.9%) was spent over sand ripples, the remaining time (34.1%) the bird foraged over flat ground. Brittle stars and anthozoans were present in most areas visited by the penguin with the former being present in 22.5 min (75%) of the benthic video footage while the latter occur for a total of 17.9 min (60%). Horse mussels were present for a total of 9.3 min (31%) of the bottom time.

Prey encounters were associated with certain benthic habitat types. All prey encounters occurred over coarse sand although the sediment structure differed depending on prey species. Opalfish were principally encountered on sediment ripples (93.6% of the total prey pursuit time, https://vimeo.com/179414724), while flat bottom habitat played a more important role during blue cod pursuits (52.8% of blue cod pursuit time, https://vimeo.com/179414724#t=2m32s). With regards to epibenthic characteristics, brittle stars and anemones were present during the majority of the prey pursuit times for both fish species (Fig. 3). However, horse mussels were present only during blue cod pursuits (81.4% of blue cod pursuit time).

### Flipper movements

When descending to the sea floor the penguin propelled itself with fast, strong flipper strokes that got progressively slower and less pronounced with time and, thus, increasing

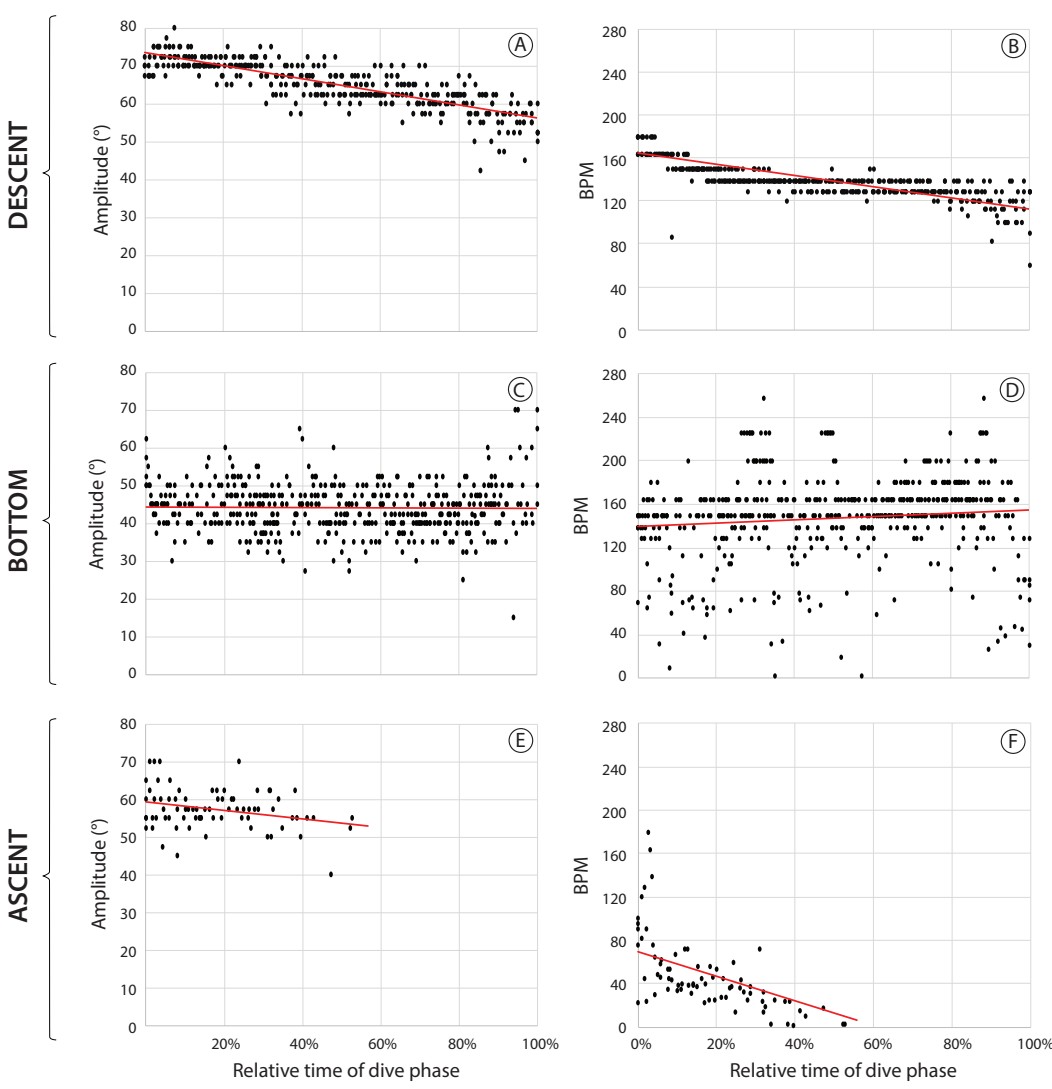

**Figure 4** **Flipper movements in a yellow-eyed penguin during the descent, bottom and ascent phases of 12 randomly chosen benthic dives.** A, C, E graphs showing flipper amplitude (i.e., maximum angle), B, D, F, graphs showing changes in flipper beat frequencies. Red lines indicate regression of the corresponding data (see 'Results' for details). Note that $x$-axis shows relative durations of the dive phases to account for dive dependent time variations.

depth (Pearson correlation – flipper amplitude: $R^2 = 0.69$, $F_{1,363} = 791.8$, $p < 0.001$, BPM: $R^2 = 0.13$, $F_{1,363} = 55.2$, $p < 0.001$, Figs. 4A & 4B; https://vimeo.com/179414575). In contrast, ascending was principally passive with the penguin using its natural buoyancy to return to the surface, occasionally aided by a few strokes in the early stages of the ascent, decreasing beat frequency (flipper amplitude: $R^2 = 0.01$, $F_{1,74} = 0.49$, $p = 0.49$; BPM: $R^2 = 0.27$, $F_{1,76} = 28.5$, $p < 0.001$, Figs. 4C & 4D) and no observable flipper movements towards the end of the dive (https://vimeo.com/179414575#t=1m49s). Despite differences in flipper movement between the two transit phases of a dive, the vertical velocities recorded

by the TDR did not differ significantly (mean descent velocity: $1.45 \pm 0.28$ m/s, mean ascent velocity: $1.36 \pm 0.57$ m/s, $n = 159$ dives, Welch's $t$-test: $t_{232} = 1.73$, $p = 0.09$).

During the bottom phase flipper amplitudes showed no correlation with relative bottom time (flipper amplitude: $R^2 = 0.001$, $F_{1,479} = 0.67$, $p = 0.42$ (Fig. 4C), likely owing to the fact that bottom phases consisted of a mix of searching behaviour and high-speed prey pursuit (https://vimeo.com/179414575#t=0m33s). While searching the penguin showed lower flipper beat frequencies ($133 \pm 48$ BPM, $n = 809$) paired with greater flipper amplitudes ($53° \pm 14°$ ) when compared to prey pursuit (BPM: $162 \pm 44$, $n = 113$, Welch's $t$-test: $t_{232} = -13.4$, $p < 0.001$; amplitude: $45° \pm 7°$ , $t_{152} = 6.4$, $p < 0.001$). Flipper beat frequency increased slightly but consistently towards the end of the bottom phase (BPM: $R^2 = 0.02$, $F_{1,484} = 8.2$, $p = 0.004$, Fig. 4D), most likely as a result of the penguin often starting its ascent back to the surface not long after successful prey captures (https://vimeo.com/179414575#t=1m45s).

### Surface breathing & underwater exhalation

Frame counts of the video footage during 12 random selected surface periods between dives showed that the penguin lifted its head out of the water to breathe for only brief moments (average duration: $0.77 \pm 0.22$ s, $n = 193$); for the majority of the time at the surface the bird kept its head under water ($1.53 \pm 1.19$ s, $n = 182$) (https://vimeo.com/179414575#t=2m25s). Duration of breathing intervals increased with ongoing duration of the surface period (Pearson correlation: $= 0.45$, $F_{1,191} = 47.4$, $p < 0.001$) indicating increased respiration activity in preparation for the next dive (Fig. 5).

During the dive, exhalation regularly occurred at the onset of phases with increased acceleration (i.e., prey pursuit). Such exhalations were brief but performed with substantial force; air was jetted from the nostrils as a fine gas spurt (https://vimeo.com/179418254). During the passive phase of the ascent, the penguin frequently exhaled as indicated by a stream of large bubbles released from the nostrils. The bird released substantial amounts of air on the last few meters immediately prior to reaching the surface (https://vimeo.com/179414575#t=2m18s). While some of this air may have been released from the plumage (c.f. *Davenport et al., 2011*) bubbles seem principally to originate from the frontal head region; there was no visible major gas release from the penguin's back region.

## DISCUSSION

The high-quality video footage provided a substantial amount of new insights into the foraging behaviour of yellow-eyed penguins and their benthic habitat. While it is impossible to draw far-ranging conclusions from only a single deployment, it nevertheless highlights that high-definition cameras provide a new tool facilitating the examination of various aspects of the foraging ecology of marine predators through direct observation. It can be particularly useful to verify and calibrate behaviours measured with other types of devices such as TDR and accelerometers.

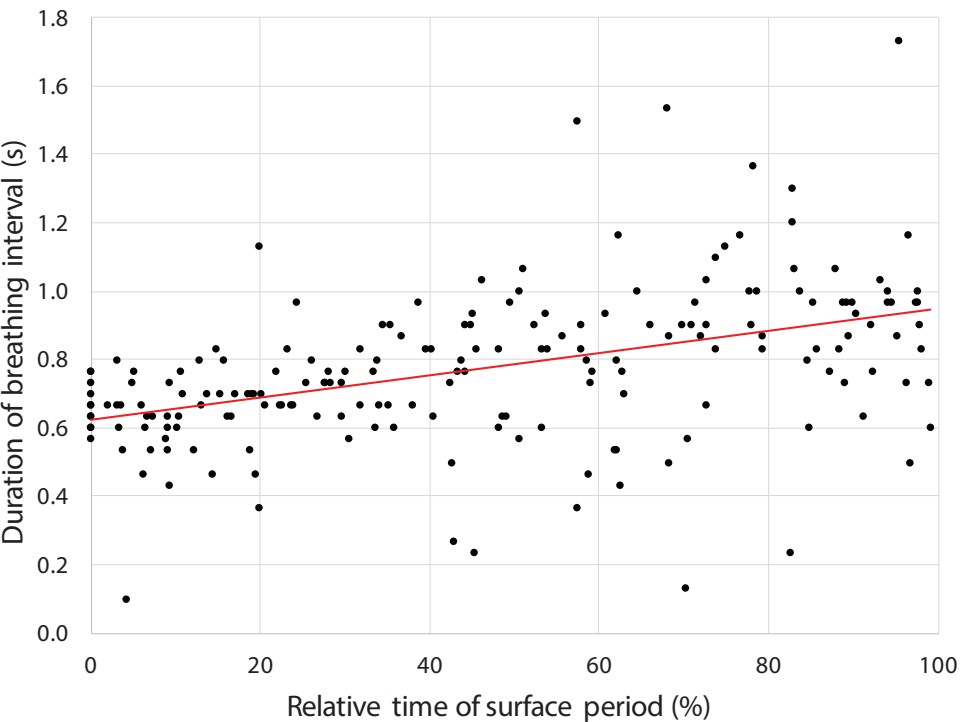

**Figure 5** **Increasing duration of breathing intervals ($n = 193$) during the surface period after 12 randomly selected dives performed by a yellow-eyed penguin.** Note that the $x$-axis shows relative time to account for varying surface period durations. Red line indicates regression of data (see Results for statistical analysis).

## Device effects

Attaching external recording devices to diving animals always comes at the cost of compromising their streamlined body shape (e.g., *Ludynia et al., 2012*), a problem that can be mitigated via device shape, size and attachment position (*Bannasch, Wilson & Culik, 1994*). At the surface there were no indications that the penguin was negatively affected by the device; the bird did not exhibit balancing problems which externally attached devices can cause in smaller species (*Chiaradia et al., 2005*), nor did it peck at the device frequently which suggests aberrant behaviour (*Wilson & Wilson, 1989*). Moreover, the number of successful prey captures further suggests that the bird's foraging capabilities were not drastically affected by the video logger. With the exception of two unsuccessful blue cod encounters, all events classified as prey pursuit were merely accelerations that did not end in any obvious prey encounter. The bird was one of the few breeders that raised two chicks to fledging in an otherwise poor breeding season.

## Predator–prey interactions & prey species importance

In line with previous descriptions of yellow-eyed penguins as primarily benthic foragers (*Mattern et al., 2007*), the penguin's prey pursuit and captures recorded during the camera operation indeed all occurred at the sea floor. Swimming very close to the seafloor could serve several purposes. It could be a strategy to flush out benthic prey that blends in with
the substrate, but it could also mean the penguin has a greater chance to see its prey from the side, and thus reduce the effect of prey camouflage. Opalfish, for example, are very well camouflaged and very difficult to make out from above (*Roberts, Stewart & Struthers, 2015*). This species seems to principally rely on its camouflage as means of predator avoidance since none of the opalfish captures involved a chase. In contrast, during both successful blue cod encounters, extended high-speed chases ensued before the fish was ultimately captured. Blue cod and opalfish differ significantly in their anatomy with the small, slender opalfish presumably lacking the physical prowess for prolonged swimming when compared to muscular blue cod (*Roberts, Stewart & Struthers, 2015*). When facing an air breathing predator, the latter strategy is likely advantageous as the predator's increased energy requirements for pursuit make escape a more likely outcome for the prey. The penguin's hasty ascent and subsequent failure to consume a blue cod it captured after a 22-second-long chase demonstrates the efficacy of this evasion strategy.

Both opalfish and blue cod have previously been found to be among the most important prey items in the yellow-eyed penguin's diet (*van Heezik, 1990a*; *Moore & Wakelin, 1997*). While both fish species have comparable energetic values (~20 kJ g$^{-1}$, *Browne et al., 2011*), the body mass of opalfish is considerably lower when compared to blue cod (*van Heezik, 1990a*; *van Heezik, 1990b*). It is possible that the energy gain from catching blue cod justifies the expenditure to catch it, while the easier-to-catch opalfish might need to be caught in larger quantities. However, recent studies suggest that blue cod might be suboptimal prey for chick-rearing yellow-eyed penguins due to their size (*Browne et al., 2011*; *Mattern et al., 2013*) so that the penguin's ability to locate prey such as opalfish might be a decisive factor with regards to reproductive success.

## Benthic environment

Judging from the total time the bird spent over a benthic environment dominated by coarse sand and sediment ripples (65.9% of total bottom time) as well as almost exclusive encounters of opalfish over such habitat (Figs 1 & 3), it can be assumed that the penguin focussed principally on this species. Blue cod encounters were associated with the presence of horse mussels. These large bivalves protrude from the seafloor and provide hard substrate for other epibenthic taxa, thereby increasing local benthic biodiversity (*Cummings et al., 1998*). Benthic habitat with increased benthic biodiversity is generally more attractive to a variety of benthic fish species, most likely due to enhanced feeding conditions (*Cranfield et al., 2001*). Our video data also suggests that the fish use the bivalves and associated cavities as shelter to avoid capture (https://vimeo.com/179414777).

The majority of prey pursuits occurred in areas that featured anthozoans, principally sea anemones (Figs 1 & 3). Anemones are known to play an important role as refugia and feeding habitats for small fish (*Elliott, 1992*) and could therefore be another indicator for locally increased biodiversity. Brittle stars on the other hand, although equally abundant, seemed to be of lesser relevance with regards to prey encounters. So, it appears that examining the composition of the benthic habitat alone might enable assessment of which prey types penguins are foraging for, though more data are required before conclusions can be drawn. However, this already hints at the potential for wide-ranging habitat analysis of

at-sea movements in benthic predators, provided that spatial distribution of the different benthic habitats can be obtained. While in our specific case, no such habitat maps exist, planned further deployments of video loggers are expected to provide the necessary environmental information.

Deploying video loggers on penguins could enable detailed mapping of the benthic habitat within the species' home ranges. Yellow-eyed penguins are known to have preferred individual foraging areas often with little overlap between birds (*Moore, 1999*). Moreover, the birds tend to often dive along the seafloor when swimming towards their foraging grounds (*Mattern et al., 2007*) so that camera logger data in combination with GPS information can be used to establish spatial biodiversity indices and benthic habitat maps.

The outer ranges of the marine habitat of yellow-eyed penguins from the Otago Peninsula is subject to bottom fisheries which have a profound effect on benthic ecosystems (e.g., *Hinz, Prieto & Kaiser, 2009*; *Queirós et al., 2006*; *Schratzberger & Jennings, 2002*). Yellow-eyed penguins have been found to forage in the wake of trawl fisheries, potentially to the detriment of their reproductive success (*Mattern et al., 2013*). Changes in sediment structure and epibenthic biodiversity as a result of bottom trawl disturbance likely negatively affect the penguins' foraging success (*Browne et al., 2011*). Camera loggers can help to determine how much of the penguins' foraging habitat has been compromised by fishing activities and what the consequences are for this species' foraging behaviour and success.

Beyond investigations of behaviour in a wider environmental context, our study also shows the potential application of camera loggers for the investigation of physiological aspects of marine animals.

## Flipper movements

Our observations of flipper movements, i.e., strong flipper movements at the beginning of a dive that decrease with depth, and cessation of flipper movements during ascent, align with findings reported in other penguins. Using accelerometers, *Sato et al. (2002)* found that King penguins showed vigorous flipper beating at the beginning of a dive to counter positive buoyancy. With increasing depth, air volume in the penguin's body becomes compressed, reducing its buoyancy so that fewer flipper beats are required. That this also applies to flipper amplitude (Fig. 4) was not detectable by using body acceleration as the only measure. A more elaborate system of sensors and magnets attached to flippers was used on Magellanic penguins which allowed the recording of both flipper amplitudes and beat frequencies (*Wilson & Liebsch, 2003*). However, the system is known to be prone to failure, rendering the use of back-mounted wide-angle cameras a much more reliable alternative. Flipper beat frequencies and amplitudes are directly related to energy expenditure (*Kooyman & Ponganis, 1998*; *Sato et al., 2011*). They provide the means for the quantification of energy budgets (*Wilson & Liebsch, 2003*) and subsequently can be used to assess individual fitness in relation to foraging success and subsequent reproductive performance (*Kooyman & Ponganis, 1998*).

We provide evidence that the ascent phase in penguins is largely passive, as has been suggested using both accelerometers and magnets (*Sato et al., 2002*; *Wilson & Liebsch, 2003*).

*Sato et al. (2002)* concluded that during ascent penguins benefit from expanding air volume in their body which increases their buoyancy as they get closer to the surface. Penguins also actively slow down their ascent and it was argued that this could be achieved by increasing the attack angles of their flippers to increase drag (*Sato et al., 2002*). Judging from body movements apparent in the video data during the ascent phases we suggest that the yellow-eyed penguin indeed adjusted flipper attack angles while ascending, although this seems to be more for steering. Based on the video footage it appears that the bird uses controlled exhalation towards the end of the ascent to control speed (https://vimeo.com/179414575#t=2m18s).

### Respiration

The video data provide new insights into the respiration of yellow-eyed penguins. To date it was unclear whether penguins exhale regularly while diving. Various studies estimated diving air volume via a penguin's buoyancy calculated from its ascent speeds at the ends of dives (*Sato et al., 2002*; *Sato et al., 2011*). However, the accuracy of this approach is compromised if the penguins were to exhale prior to their final ascent (*Ponganis, St Leger & Scadeng, 2015*). The video data clearly showed that the penguin generally exhaled when accelerating during prey pursuit so that models estimating diving air volume via the proxy buoyancy must take acceleration into account. The fact that the penguin exhaled when accelerating probably serves the purpose of reducing blood $CO_2$ and mobilizing $O_2$ from oxygen stores for prey pursuit. Such pursuits must be costly in terms of oxygen consumption as is evident from the observed consecutive prey encounters during one single dive, which resulted in the penguin letting go of the second fish after a rapid ascent to the surface (https://vimeo.com/179414724#t=3m07s). Unlike seals that have been found to exhale when ascending from deep dives, most likely to reduce the drop in blood oxygen (*Hooker et al., 2005*), the penguin principally exhaled during the second half of the ascent possibly indicating adjustment of buoyancy and ascent speed (but see also *Davenport et al., 2011*). Reoxygination during the surface period in penguins is highly optimized (*Wilson et al., 2003*). Inhalation events at the surface are brief so that the bird can frequently lower its head into the water, presumably in an effort to look out for potential predators (e.g. sharks, sea lions; *Seddon, Ellenberg & van Heezik, 2013*). Extensive exhalation prior to resurfacing also prevents pulmonary barotrauma and facilitates immediate inhalation once back at the surface.

## CONCLUSIONS

The deployment of a full HD video logger on a yellow-eyed penguin resulted in a versatile visual data set that provided a variety of information well beyond what was initially intended. Enhanced video quality allows detailed analysis of the benthic environment as well as prey encounter rates and prey composition. In combination with GPS data, the potential for a comprehensive survey of benthic ecosystems is substantial highlighting the multi-disciplinary potential of such data.

A large field of view achieved through wide-angle lenses furthermore allows detailed analysis of flipper movements, which to date could only be achieved through elaborate

modelling of accelerometer data (*Sato et al., 2002*; *Sato et al., 2011*) or use of complicated magnetic logger setups (*Wilson & Liebsch, 2003*). Neither of these setups provided information about exhalation, which appears to play a much more important role during diving than previously thought. When comparing video data recorded here with videos from previously published studies (e.g., *Watanabe & Takahashi, 2013*, https://vimeo.com/268905870) it becomes clear that greater visual fidelity of full HD cameras comes along with a much wider range of quantifiable data. This creates a new opportunity for a more holistic approach to study the diving behaviour of marine animals that integrates behaviour, physiology and their environment.

Depending on which behaviours are quantified, the manual analysis of video data can be quite time-consuming. For example, flipper beats and angles require a frame-by-frame analysis; an average dive duration of 3 min translated to 5,400 frames per dive. However, the higher the resolution and quality of the video footage, the greater the potential to develop machine learning algorithms (such as Google Cloud Video Intelligence; https://cloud.google.com/video-intelligence/) that may be used to automate the analysis process. For more basic analyses such as prey composition and encounter rates, but also determination of environmental parameters, there already exist software solutions that offer an enhanced workflow, for example the video annotation software BORIS (http://www.boris.unito.it/).

Obviously, there are still limitations to the use of camera loggers. Restrictions arise from the battery life as well as the memory to store high definition video data. In our case, 15 min of footage resulted in video file sizes of 1.5 gigabytes. Moreover, the deployment with the camera set-up we used requires a certain amount of predictability, particularly knowledge about how soon after departure the bird is likely to engage in behaviours that are of interest (e.g., prey pursuit). For all these reasons, the technology currently available is best suited for short-term deployments on central place foragers. Although video data recorded on animals performing long-term foraging trips (e.g., Magellanic penguins, *Boersma & Rebstock, 2009*) might still deliver valuable data, this has to be weighed against the fact that external devices inevitably have an effect on the animal's foraging ability (*Bannasch, Wilson & Culik, 1994*; *Ludynia et al., 2012*). This could be alleviated by incorporating further mechanisms to control camera recording (e.g., duty-cycling of recording function, pressure control). While the use of animal-borne cameras for scientific research is still in its early days, the enormous potential of this technology will doubtlessly result in devices incorporating more elaborate functionality in the future.

## ACKNOWLEDGEMENTS

We would like to thank Horst Mattern, Melanie Young and Jim Watts for help in the field, and Leon Berard for first preliminary evaluation of the video data. Special thanks are due to Bruce McKinlay (Department of Conservation) for supporting this project and facilitating the permitting process of a novel bio-logging method.

### Funding
This work was supported by an Otago Research Grant [PL 112034.01 R.FZ] issued to Philipp J. Seddon. The funders had no role in study design, data collection and analysis, decision to publish, or preparation of the manuscript.

### Grant Disclosures
The following grant information was disclosed by the authors:
Otago Research Grant: PL 112034.01 R.FZ.

### Competing Interests
Yolanda van Heezik is an Academic Editor for PeerJ. Michael D. McPherson is director of CTNova Ltd.

### Author Contributions
- Thomas Mattern conceived and designed the experiments, performed the experiments, analyzed the data, contributed reagents/materials/analysis tools, prepared figures and/or tables, authored or reviewed drafts of the paper, approved the final draft.
- Michael D McPherson conceived and designed the experiments, contributed reagents/materials/analysis tools, authored or reviewed drafts of the paper, approved the final draft.
- Ursula Ellenberg conceived and designed the experiments, performed the experiments, contributed reagents/materials/analysis tools, authored or reviewed drafts of the paper, approved the final draft.
- Yolanda van Heezik conceived and designed the experiments, contributed reagents/materials/analysis tools, authored or reviewed drafts of the paper, approved the final draft.
- Philipp J. Seddon conceived and designed the experiments, contributed reagents/-materials/analysis tools, authored or reviewed drafts of the paper, approved the final draft.

### Animal Ethics
The following information was supplied relating to ethical approvals (i.e., approving body and any reference numbers):
The University of Otago's Animal Ethics Committee approved this study (UOO AEC 69/15).

### Field Study Permissions
The following information was supplied relating to field study approvals (i.e., approving body and any reference numbers):
Filed experiments were approved by the New Zealand Department of Conservation (45799-FAU).

## Data Availability

The full and unaltered video has been made available publicly on YouTube and can be accessed at https://youtu.be/wKuFsGlfS8A.

## Supplemental Information

Supplemental information for this article can be found online at http://dx.doi.org/10.7717/peerj.5459#supplemental-information.

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
