# Peer review of "High definition video loggers provide new insights into behaviour, physiology, and the oceanic habitat of a marine predator, the yellow-eyed penguin"

_PeerJ, doi:10.7717/peerj.5459_

## Round 0.1 · original submission · Major Revisions

Editorial decision: Major revisions

I agree with both reviewers that the paper needs more citations. Citations about the benthic habitats, as suggested by Reviewer 1, would be useful, but if they don’t exist, I don’t think that would be a fatal flaw in the paper. Please think carefully about the goal or goals of the manuscript and make sure that it is clear and is addressed by the results and conclusions. A more detailed comparison between your video logger and loggers used in previous studies, as suggested by Reviewer 1, would be helpful. The comparison should include both the specifications of the video loggers and the results. Please discuss the representativeness of your one deployment. I disagree with Reviewer 1 about dropping the second goal. I think the behavior recorded is more interesting than the technical details of the logger (although that is important to your first goal). I also disagree with Reviewer 1 about your interpretations of the video and the need for GPS and TDR data to correctly interpret the data. I do agree with Reviewer 2 that most of the behavior can be detected using other types of loggers, but I would much rather tape a video logger to a penguin’s back than try to attach inter-mandibular sensors!

Editor’s review:

The paper is well written. I like that you addressed some of the unexpected behaviors evident in the videos. I have a few comments and suggestions, and included some editorial suggestions in the pdf.
I agree with David Ainley that penguins aren’t really top predators. They’re preyed on by sharks, orcas, sea lions, leopard seals, even sometimes giant petrels, depending on where they occur. They are not at the tops of their food webs.

Methods:
This section is mostly clear and well written, but needs more explanation on a few points. It also needs some explanation of the statistics and selection of random sections.
Line 115: What are alarms? What alarms were programmed? Why?
Line 131: What is video scrubbing?
Lines 161-162: Did the GPS logger not work?
Say a little about what statistics you did. It’s not always clear when you present statistics in Results what you were testing and why.
Explain that you used 10 random sections of video for several analyses, and justify why you chose 10. Were the random sections selected from the entire 2 hours 8 seconds, or were they chosen from the sections of video that recorded the relevant behaviors only?

Results:
Again, this section is mostly clear, but needs to match the order of Methods.
Subheadings should be in the same order as in Methods.
Line 186: 09:00-16:10 hrs were devoted to benthic diving – except the last 10 minutes? Maybe should say 09:00-16:00 hrs.
Line 187: How often were frames lost? Any pattern to the loss?
Line 202: Unclear, please re-word.
Several of the links to videos where a specific time is targeted don’t work.
Line 231: I think you mean Figure 2.
Lines 245-247: You say ‘no correlation’ but report p < 0.001 for flipper BPM. Please double check the statistics you report.

Discussion:
Make sure the order is consistent with Methods and Results.
Flipper movements: Cite Wilson et al. 2010 Aquatic Biology.
Line 369: There might be other reasons to exhale towards the end of the ascent, such as preparing to inhale as soon as possible. Human scuba divers exhale during ascent to prevent gas expansion in the lungs. Could this apply to penguins?
Lines 379-380: citations?
How useful is this set-up for penguins that make long foraging trips? Magellanic penguins, for example, are gone for 2-4 weeks during incubation, and ~1 week when chicks are older. Would it be worth it to put such a large device on a penguin to capture data for a few hours out of a week-long or longer trip? Could the camera be programmed to turn on below a certain depth? This might be worth some discussion.
It might also be useful to say something about how many frames of video were collected and how much time it takes to get useful information from the video. What process did you use to isolate interesting behaviors?

References:
There are some font issues and missing information in references, e.g., Moore et al., Volpov et al. Please double-check all references.

Figure 1:
The caption and y-axis label say proportion, but the y-axis scale is labeled as percentages. Please choose one and be consistent.
One darker and one lighter color would make this print better in black and white.

Figure 2:
This figure seems to have a lot of information in it, but is confusing. Please explain it better in the caption. Print it in black and white (as some of your readers will) and see if it makes any sense. I agree with Reviewer 1 that use of symbols instead of colors could improve the readability of this figure.

Figure 4:
Please make axis scales and labels big enough to read. Using 3 rows by 2 columns might help.
Be careful extrapolating regression lines beyond the data, as in E.

Reviewer 1 ·

Basic reporting

This article was written in clear and unambiguous, professional English. Throughout a whole manuscript, the sentences were easy to read and understand. One issue in terminology is ‘foraging ground’ in line 123. To my knowledge, foraging ground means a spatial area for foraging of a species having high fidelity on the site (Casale et al. 2012). In line 123, both of the use of term and meaning of the sentence is ambiguous. Did the penguin have another foraging ground on land? I would recommend to change the term into more general terms such as land or breeding site, or authors can introduce the definition of ‘foraging ground.’ I found the reference for line 123 (Mattern et al. 2007), but could not find enough information to understand the sentence, line 123. Other minor issues of terminological ambiguity could be addressed after significant changes in content of this manuscript.
One of major issues of this manuscript is reference. Many ideas and interpretations were not sufficiently supported by appropriate references. To show that the presented study has a significant difference in the angle of view, in line 111, the specification of the lens should be reported, and compared to the specifications of lens used by other studies. In the part of ‘Analysis of behaviours’, criteria for the behavior categories (prey pursuits, surface breathing, and flipper movements) were not supported by previous studies. I am sure that there are many previous studies to provide supporting evidences about the criteria. In ‘Analysis of benthic habitat’, the classification methods (sediment type, sediment structure, and composition of the epibenthic communities) should be justified by appropriate references. Especially, I would recommend to add enough number of references to explain the choice of species for habitat description. If authors can present data about the local marine biodiversity or species composition, the analysis of benthic habitat can be improved in the reliability. In discussion, line 372-373 needs supports from references.
Through a supplementary data, raw data was shared, but the dataset does not include data from other loggers, GPS and accelerometer. In line 102-104, authors reported that the other loggers were also deployed on their study animal, but the data was not presented in the manuscript and raw data. For further discussion, those data need to be presented.
Figure 2 presented important information for the discussion part of this manuscript. I agree with the design of the figure is good for showing the general pattern, but would recommend to use symbols instead of colorations for better visibility to various types of readers (i.e. color blind). Statistical values should be presented in Figure 3 & 4. Additionally, the line and axis titles of figure 3 & 4 would be better to see in black rather than gray color.
If I understand well, the main goal of this manuscript is to present potential usefulness of animal-borne video camera. I agree that the authors made the best use of the video data ranging from prey pursuit to flipper movements. However, without a specific goal or prediction, it is hard to tell that the observations addressed a significant progress in the field of bio-logging. Already, many researchers has investigated behaviors of various marine animals including penguins using animal-borne video camera. Moreover, the results about flipper movements and respiration and the interpretations are ambiguous and doubtful because of the limitation in sample size, recording coverage, and supporting evidences. Also, the discussion about benthic environment could be elaborated by having GPS data from presented study and references about the local marine environment and/or species diversity. Thus, I would say that this manuscript is not ‘self-contained’ enough for publication.

Experimental design

To my understanding, the major goals of this manuscript were (1) introducing technological advances in animal-borne camera, and (2) to present the best use of the improved technique. Unfortunately, this manuscript could not address neither of the goals very well. The first goal has not been well presented in this manuscript because of the lack of enough references to compare with. For instance, authors may be able to compare the resolution, angle of view, or signal-to-noise ration of their video data with previous studies using shared supplementary data such as Choi et al. (2017).
A major hindrance for the second goal is a limited sample size and recorded time period. In spite of the limitation, it may have a significance to present the result as a potential measurements of behavioral analysis using animal-borne camera such as flipper movements. However, the new approaches in this manuscript did not have sufficient supports from previous studies, and failed to provide a new insight into the behavior of marine predators.
For further submission, I would recommend to change the focus of this manuscript into a comparison among various animal-borne camera techniques from current two-aim structure. To achieve the goal, it will be required to have data about specification of various animal-borne camera system including angle of view, resolution, signal-to-noise, and potential products from the various techniques. If authors want to keep the second goal of the presented manuscript, GPS and TDR data will be critical for providing reliable results.

Validity of the findings

I would say that more rigorous comparison with previous techniques will be necessary to show the novelty of the presented study. With current version, it is hard to tell any significance and novelty of the findings. Also, I would highly recommend to include data from GPS and TDR for the robustness of the results. Some of the results are overstated in discussion section. Especially, the part of breathing and exhalation have only weak supports from the results. For instance, only with the video, it is really hard to tell if the penguin actually breath at the time. Also, the air bubble during foraging can be generated by the fast speed of a focal individual, and the size of bubble could be increased during ascending periods due to the reduced water pressure. The limited data from one individual could not rule alternative explanations out effectively.

Reviewer 2 ·

Basic reporting

In the ms entitled “High definition video loggers provide new insights into behavior, physiology ,and the oceanographic habitat of marine top predators” the authors present how the employment of a camera logger that records full HD and has wide angle lens can shed light on the foraging behavior of the Yellow-eyed penguin. The manuscript is clear and well written however some of the presented results can also be obtained by means of other electronic devices (e.g. diving behavior, flipper movements, breathing events). By reading the objective and the last sentence of the abstract I expected authors to concentrate on presenting the new information that this video logger allows to get and not giving so much detail about data that can be obtained by means of other devices. The paper is well organized however too long, the advantages of using this new device can be express in a shorter and concise way. All along the introduction and discussion authors should state more clearly the advantages of this new logger, the possibilities it offers to improve our understanding of not only the Yellow-eyed penguin behavior but also other animals (which is the main objective of the work). For example, wide angle lens could improve our understanding of how penguins select and follow fish within a shoal and how they interact with other penguins or other species. Another drawback the work has is that the presented results were obtained from only one animal. This limits the strength of general conclusions. Figures as well as the raw data supplied are clear.

Experimental design

The objective of the manuscript is within the scope of the journal. In objective presented in the abstract (which is slightly different to the one presented in the introduction) is too general. In the abstract authors state that the main objective was the analysis of the foraging behavior. Foraging behavior includes many aspects and authors only concentrate in one; the prey pursuit behavior. I suggest modifying the objective and present it in the way it is at the end of the introduction.

Validity of the findings

As I stated before, findings are interesting but many of them can be obtained by means of other electronics devices. For example, diving parameters (depth, time spent in the different diving phases, velocities) can be obtained by means of deploying TDRs alone. Flipper frequency data can be obtained by means of using accelerometers and breath frequency can be obtained by means of inter-mandibular sensors. Authors should make more emphasis on the advantages this new logger has in comparison to other ones and on the advantages of having depth, video at high definition and wide-angle lens all together. They should also make more clear that this information was obtained from only one animal and present the limitations this has.

Additional comments

Introduction

Line 56: More references can be incorporated here together with Takahashi et al. 2008. For example Pongalis et al. 2000. JEB 203
Line 57: Thiebot et al. 2017 not "in review"
Lines 68-70: Please be more specific with these statements. Which aspects of behaviour? Which are the new opportunities for visual analysis does this new device with high definition and wide angle lens offer?
Line 73: Here yellow eyed penguin is not capitalized while in other places of the ms it is. Please keep the same format all along the text.
Line 90: Potential of both, full HD and wide angle optics not only full HD.

Materials and methods

Please mention the permits under which this work was performed.
Line 103: AXY depth devices from technosmart also record acceleration data and this is not mentioned in the methods. Please include this.
Line 109: How much time did the instrumentation procedure last? How was the animal captured?
Line 161: Authors say "future deplyments with a functional GPS". In line 104 they stated that the high definition logger had a GPS. Was it not functional? Please clarify this.
Line 171: Which diving characteristics were used to clasify benthic and pelagic dives?

Discussion

Line 270: Authors say that the device did not appear to substancially affect the penguin´s mobility. How do you know this? It would be interesting to compare this penguin mobility with the mobility of another individual with a smaller device.
Line 279-280: How do authors know that the number of successful prey captures where not affected by the logger? Do you have and estimation of how many successfull captpures these penguins perform per foraging trip in order to make a comparison?

---

## Round 0.2 · Minor Revisions

The manuscript has been markedly improved. The two reviewers and I now have only minor comments. In addition to the suggestions below, see wording suggestions in the pdf file (peerj-23829-mattern_et_al_REV1-EditorComments.pdf).

Please change ‘Yellow-eyed penguin’ to ‘yellow-eyed penguin’ throughout, except when it is at the beginning of a sentence. The usual convention is to capitalize common names of birds (both words, e.g., Yellow-eyed Penguin) in bird journals but not in other journals. The first word, but not the second word in the name, is always capitalized if it is a proper noun, such as Snares penguin or Adélie penguin. Yellow-eyed penguin does not qualify.

Lines 179-180: This sentence does not belong under ‘Benthic habitat’. I understand that it’s meant to be a bridge to the next sections, but I don’t think it’s necessary here. Please move it or delete it.

Line 277: And during unsuccessful pursuits, or were these also blue cod pursuits?

Lines 402-404: Please move or delete, same as lines 179-180.
Lines 439-440: Do you have any citations for this sentence (“The fact that the penguin exhaled when accelerating probably serves the purpose of reducing blood CO2 and mobilizing O2 from oxygen stores for prey pursuit.”)?

Line 497: It’s customary to thank the anonymous reviewers.

Here’s the full citation for the Wilson et al. paper I suggested adding to the “Flipper movements” section in the Discussion, in the first round of reviews:

Wilson RP, Shepard ELC, Gómez Laich A, Frere E, Quintana F (2010) Pedalling downhill and freewheeling up; a penguin perspective on foraging. Aquatic Biology 8:193-202. DOI: 10.3354/ab00230.

The PeerJ office informed me that you can use the video from Watanabe & Takahashi (2013) as long as you “fully cite the original source”. The text of the video caption should include the original article citation, including authors, DOI, etc. The PeerJ production group will help make it clear that the video remains under its original copyright (publication in PeerJ doesn’t make it open access).

Reviewer 1 ·

Basic reporting

No comments

Experimental design

No comments

Validity of the findings

No comments

Additional comments

The authors have substantially reworked their manuscript and the revised version make the goals clear than the previous version. By the addition of the video comparing the quality of videos from current and previous animal-borne cameras, I feel that my previous concerns about the significance of the study have been addressed adequately.

In the previous review, I suggested to use GPS data for further habitat assessment, and the authors discussed the unavailability of GPS data by a technical issue. I understand that the technical deficiency is inevitable at the very first step to address a new technique. I hope to see the further investigation with proper GPS information and land coverage data of the habitat in near future.

After the substantial revision, this manuscript is well written, and adequately showing the potentiality of using the HD and wide-angle video logger to investigate the underwater behavior of marine animals. In general, I agree that this manuscript is prepared to publish, but I have few minor comments before the acceptance.

Minor comments
Line 222-223: The video for comparing video quality between current study and Watanabe and Takahashi (2013) may be better to include the information about water depth or other factors affecting light intensity. In addition, DVL-100 is not the most recent version of Little-Leonardo video camera logger, the authors may want to compare the Mobius action-cam logger with the most recent models from other makes.
Line 243: The URL for the video does not work. If it is the part of video 4, writing the time point will be enough.
Line 245, 251, 265, 295, 304, 417 & 430: Same problem with Line 243
Line 361-362: It was hard to find the fish using the bivalves as shelter in the video. It would be helpful for audiences to make arrows pointing the fish in the video.

Reviewer 2 ·

Basic reporting

This new version of the ms # 23829 entitled “High definition video loggers provide new insights into behavior, physiology and the oceanographic habitat of a marine predator, the Yellow-eyed penguin” improved substantially. The authors made an important effort to incorporate all the suggestions made on the previous version.

Experimental design

Authors explained in detail those issues that were not clear on the previous version. I only have a few minor comments:
1) Analysis of behaviours & habitat, line 166. It would be nice to have a definition of what a dive cycle means. It is the dive plus the pre-dive surface interval? Plus the post-dive surface interval?
2) Surface breathing and underwater exhalation, line 218. “Described” instead of descried
3) Dive data analysis, line 231. Which R version did you use? Please include this piece of information.
4) Methods, line 158. It would be interesting to incorporate a sentence saying that the instrumented bird continued breeding normally after the deployment (now this information is in the discussion, line 380-381).

Validity of the findings

The ms has the drawback that the information comes from the instrumentation of only one animal; however authors have now included a section that mentions this limitation at the beginning of the discussion.

---

## Round 0.3 · Minor Revisions

The manuscript is very close. We're down to very small and picky edits. Please see the edits I made with track changes, and also pay attention to the comments in the pdf.

What happened to the figure captions? I think they were missing in the last version also, but I forgot to go back and check.

References:

Please italicize scientific names of species.
Some of the DOIs contain “http:\\dx.doi.org\” and others don’t. Please be consistent.
BirdLife: what is the string of characters at the end?
Ellenberg and Mattern 2012: is this a journal article, report, book chapter?
Mattern et al. 2007 & Mattern et al. 2013: authors’ initials repeated.
Noda et al. 2016: journal name abbreviated (no abbreviations on the rest of the references).

---

## Round 0.4 · Minor Revisions

I'm sending this back to you because of the changes needed in the figure captions. Sorry I didn't catch these in the last round (but you should have caught them too).

Please use lower-case “y” for “yellow-eyed penguin” in all figure captions.
The captions for Figures 2 and 3 are reversed, i.e., the caption with Figure 2 belongs to Figure 3, and vice versa.
Add an apostrophe to Figure 3 caption (now listed as Figure 2): “Timeline of a yellow-eyed penguin’s prey pursuit events …”.
Figure 4 caption does not match the rearranged figure. It refers to rows which are now columns.
Also, please see a few more edits in the annotated pdf.

Although the standard email from PeerJ says acceptance is not guaranteed, I consider the paper accepted once these minor edits are complete.

---

## Round 0.5 · accepted · Accept

During production, please make sure you correct the figure caption for Fig. 4. It still refers to the top and bottom rows when it should refer to the left and right columns (perhaps you uploaded the wrong version?) Also, on line 231, 'camera' should be 'cameras', and on line 313, 'be' should be 'by'.

#